# Detection of the Sinking State of Liquid Oil in Breaking Waves Based on Synthesized Data: A Behavior Process Study of Sunken and Submerged Oil

**Shibiao Fang** [1,2,3] 🆔 **, Lin Mu** [1,2,3,*] **, Kuan Liu** [1] **and Darong Liu** [4]

1. College of Computer Science and Software Engineering, Shenzhen University, Shenzhen 518060, China; wuyibiaobiao@163.com (S.F.); kuanliu0223@gmail.com (K.L.)
2. College of Life Sciences and Oceanography, Shenzhen University, Shenzhen 518060, China
3. Southern Marine Science and Engineering Guangdong Laboratory (Guangzhou), Guangzhou 511458, China
4. College of Marine Science and Technology, China University of Geosciences, 388 Lumo Road, Wuhan 430074, China; lidr1169@cug.edu.cn
* Correspondence: mulin@szu.edu.cn

**Abstract:** In computer vision, pollutant detection is a highly concerning issue, and it has been widely used in the fields of pollutant identification, tracking, and precise positioning. In the ocean, oil tends to disperse into the water column as droplets under breaking waves, and it is called sunken and submerged oil. Aiming at the most difficult issue of identifying liquid submerged oil pollution, this paper proposes a method of synthesized data containing specific markers for oil detection. The Canny operator was used to remove the background of the liquid submerged oil. Then, affine transformation was applied to simulate the real situation of oil deformation. Linear mapping was presented by matrix multiplication, and translation was represented by vector addition. At last, bilinear interpolation was used to integrate the oil into the image of the laboratory pictures. In addition, this research randomly added interference information, so that the probability distribution of synthesized data was closer to the probability distribution of the real data. Then, this paper combined various methods to improve the accuracy of liquid oil detection, such as Feature Pyramid Networks, RoIAlign, difficult sample mining. Based on the above methods, 1838 images were synthesized in this paper and combined into a training set. The results show that the average accuracy of the oil detection is increased by 79.72%. The accuracy of the synthesized data method for labeled oil detection was 18.56% higher than that of oil detection without labeling. This research solves the difficulty of obtaining sunken and submerged oil images and the high cost of image annotation.

**Keywords:** oil spill; submerged liquid oil; synthesized data; laboratory environments; pollutant behavior detection

## 1. Introduction

Offshore oil spill has a great impact on marine economic development and ecological environment protection. After the oil leaks into the sea, the oil may gradually settle since the oil's density after weathering exceeds the density of sea water. Or it may combine with suspended sediment, or it may disperse under the action of dispersant [1–3]. Under breaking waves, such oil will suspend in sea water, or sink on the seabed (it will be resuspended quickly), and they are referred to as sunken and submerged oil [4]. At present, oil in breaking waves has the characteristics of unknown source, difficult detection, difficult tracking and inefficient capture and disposal [5–9]. Research on the natural dispersion of oil in the ocean shows that oil tend to disperse into the water column as droplets under breaking waves [10–14]. However, most of the experimental data available on liquid oil in the literature were obtained with simplified techniques [15]. These techniques include agitated vials or laboratory flasks [16,17], wave tank experiments [12], and plunging

jets [18–21]. The focus of many early experiments was to study the oil vertical mixing and emulsification processes. In most of these experiments, liquid oil's dispersion area and sinking depth were not measured. In recent studies, more attention has been given to the sinking state of liquid oil under breaking waves, namely, the behavior process of sunken and submerged oil. Reed et al. [20] conducted a series of experiments to measure oil size under breaking waves for different oil types at different weathering conditions. Zeinstra-Helfrich et al. [22] studied different mechanisms of liquid oil dispersion, using empirical methods to simulate the behavior process of oil under breaking waves. Johansen et al. [23] presented an improved equilibrium model (a model that calculates a single droplet size distribution), but this model cannot be used to analyze the distribution characteristics of liquid oil. To sum, the research on the mechanism of sunken and submerged oil has always been a difficult issue in the research of oil spills around the globe [24–27]. Therefore, the rapid and automatic detection of submerged oil's behavior process is not only helpful to marine pollution assessment but also important for the development of an oil monitoring and prediction model. However, there are few research cases on the detection and tracking of the behavior process of sunken and submerged oil at home and abroad [28–30].

The existing target detection usually extracts the histogram or SIFT (Scale-invariant feature transform) feature of the image, performs single mapping match on the key points, or extracts the HOG (Histogram of Oriented Gradient) feature of the image for training. In some improved methods, Liu et al. used Gaussian affine differences to locate key points [31], and constructed object prototypes by analyzing elliptical features. Redmon et al. used manual feature selection, and used gradient direction histogram pyramid to improve feature selection, and then sent it to SVM (support vector machines) for classification [32]. Some researchers proposed a fast target detection algorithm, which used the edge growth of the characteristic rectangle to solve the problems encountered in target detection [33–35]. Bochkovskiy used the Fisher vector and Hamming distance for recognition and improved the recognition effect by voting on the classification results [36]. Girshick fused multiple features based on a regression model and constructed a new index vector based on the similarity of multiple features of the image [37]. Zhang et al. used the integral map to extract the integrated channel features of the candidate area of the traffic sign on each channel of the image [38], and then used Boosting and soft cascade methods to learn and classify the features. Sun et al. proposed an iterative codeword selection algorithm to generate a discriminative codebook as the characteristics of the candidate area of traffic signs and used SVR (support vector regression) to identify traffic signs from the candidate area [39]. M. Shams et al. combined Bag-of-Word and the spatial histogram as the features of candidate regions and used SVM to classify the candidate regions. In order to improve the detection speed, others used GPU-accelerated fuzzy nearest neighbor algorithm to classify markers [40].

Traditional target detection methods usually use basic image features such as color, shape, and texture for extracting image information [41]. However, in the problem of oil recognition, the color features are too single. Then the shape of the oil is mainly regular polygons such as circles and squares, and the texture is relatively simple. In images, there may be oil deformation, partial occlusion, partial color change, etc. The overall similarity of these images may not be high, but there are also cases of target loss. In actual scenes, oil in natural scenes may appear anywhere in the image, and the size is relatively small. The oil also has various states such as translation, rotation, and diffusion. The picture will also be shaken when shooting. As such, the image quality is not good due to these reasons, and traditional image features cannot cope with these problems well, making it more difficult to detect oil.

Compared with other target detection objects, oil images in natural scenes are complex and uncertain, which makes oil detection in natural scenes more complicated and difficult than target detection in life. How to filter complex images from natural scenes with background characteristics, it is particularly important to get the oil's area for identification. Using existing recognition algorithms to detect natural scene images, the accuracy rate is

relatively low, and the recognition effect is not ideal. In recent years, deep learning has made a leap-forward breakthrough in image recognition [37]. After large-scale data sets such as ImageNet and MSCOCO are released [38–40], related algorithms are also open source [41–45]. Then, deep learning has also made great progress in the field of object recognition [46–48]. In this article, target detection based on deep learning is mainly divided into three parts: image feature extraction, candidate region selection, and candidate region recognition. The feature extraction stage is mainly to obtain the information contained in the image, including the color, texture information and high-level semantic information of the image. Compared with traditional recognition methods, the biggest advantage of deep learning is that it can automatically learn features from big data without human design features. The method based on deep learning usually contains thousands of parameters, which can better obtain the feature expression of the image and achieve better results. The commonly used method for selecting image candidate regions is sliding window, that is, a window of a fixed specific size traverses the image, and objects in the image that meet a certain size and have specific characteristics will be selected. When identifying candidate regions, no separate classifier is used, but a classification neural network is added, and the feature representation and classifier are jointly optimized to realize convolution sharing, thereby reducing hardware overhead and saving time cost of calculation. The accuracy and performance of the detector are improved.

However, most of the current deep learning detection algorithms are supervised and need to be trained using labeled data in advance [49,50]. They usually select data that can represent the actual situation. Data sets generally include public data sets and self-prepared data sets. Self-made data sets usually need to collect a certain number of representative pictures. Generally, image data of various angles and shapes are collected through methods such as crawling and manual shooting. The data are manually labeled. But for the application scenarios of liquid submerged oil detection, there is no public data set about it. Therefore, this article hopes to expand the labeled data through synthesized data in oil detection, so as to reduce the cost of labeling and improve the accuracy of oil detection. However, unlike text recognition, liquid oil may produce non-rigid deformation, reflection and occlusion. There may be great differences between the various forms of oil. Therefore, how to effectively synthesize oil images is a problem that needs to be solved urgently.

Up to now, the deep learning detection of laboratory's liquid oil under breaking waves has not been reported at home and abroad. The objective of this paper is to obtain a framework for liquid oil detection and identification, and then to detect and identify the common liquid sunken and submerged oil forms in the laboratory. After detecting the dispersion area and depth of liquid oil, this research will further use this information to monitor and track the migration of oil, and it has great practical value and application prospects.

*The Novelty of This Research*

Neural network was first proposed by Fukushima in 1980, which is the first biological neural-based network, and it is Neocognitron [51]. Lecun proposed the concept of convolution and pooling in 1989 [52], which is also the earliest realization of convolutional neural network. Then Lecun proposed the landmark of Lenet-5 [53]. The basic structure of this network is the same as the commonly used basic structure today. It is a relatively complete small convolutional neural network, including convolution layer, pooling layer, fully connected layer and other basic structures. Moreover, it lays the foundation for the subsequent development of neural networks. However, with the release of large data sets and the improvement of machine computing performance, in the ImageNet competition in 2012, AlexNet stood out, and it became the champion of the ImageNet competition classification project. The classification top-5 error rate was 10% lower than the previous one, achieving a breakthrough in history and opening a new chapter in deep learning. Since 2012, the convolutional neural network has shown a phenomenon of blooming. Researchers have

improved the network through various techniques. In order to improve the classification accuracy, they mainly focus on deepening the network depth and enhancing the function of the convolution module. These two methods improve the accuracy of the network through different principles, but at the same time, the network becomes more complex [54–56]. In this paper, VGG (2014 ImageNet competition detection task champion) is applied to solve the problem of the lack of liquid oil annotation data.

Aiming at the changing shape of oil in natural scene images, this paper proposes a method of synthesizing data containing specific markers for oil and automatically generates labeling files during computer synthesizing, so that a large amount of labeled data can be obtained quickly. In the application scenarios of detection, artificially synthesized sample training data is used. Using the Canny operator to obtain the significant changes in the brightness of the local area of the image, this research separates the foreground and the background of the image to extract the oil. Performing affine transformation on the oil, it presents a linear mapping through matrix multiplication, uses vector addition to represent the translation and simulates the original data for oil deformation. Then this paper uses the bilinear interpolation to perform a linear interpolation from the pixel level in each of the two directions in space. It adds Gaussian blur and salt and pepper noise for randomly increasing redundant interference information in the image to make the synthesized data's probability distribution closer to that of the original data. The final original set includes 2300 artificially labeled pictures, including 1908 in the training set and 392 in the test set. The synthesized data set includes 2280 pictures, of which 1838 in the training set are synthesized data, and 442 in the test set are artificially labeled data.

The oil detection method based on multi-strategy fusion is studied to improve the detection accuracy. First, the feature pyramid is used for feature fusion. By adding upsampling and pixel values, the texture features of the image are combined with high-level semantic features, which reduces the position offset caused by multiple sampling of small objects. Then it uses RoIAlign to cancel the quantization operation and uses bilinear interpolation to calculate pixel values, reducing the position information offset caused by the quantization operation in the local feature mapping. Finally, the samples with high confidence are mined, and the entropy is used as the difficulty feedback of the samples, which improves the model's judgment on difficult samples such as small objects.

## 2. Materials and Methods

The object of this study is the medium liquid oil under the action of three kinds of breaking waves. The density of each oil mass used in this research is 0.93 g/cm$^3$. The images used in this study are from the key laboratory of marine oil spill identification and damage assessment technology of the State Oceanic Administration, and they were photographed from 1 April to 28 June 2021. In the experiment, the wave breaking time has a good law because the push plate wavemaker is used to simulate the generation of sea waves. The wave phase focusing method is used to generate breaking waves. The specific principle is that the long wave propagation speed is fast and the short-wave propagation speed is slow. Then these two waves superimpose to generate wave breaking at a certain time [57]. This research changes the parameters of the wavemaker and sets the wave heights. There are several breaking waves with wave heights of 15 cm, 25 cm, 35 cm in the laboratory.

This paper combined original data and synthesized data for model training. The synthesis framework is as follows: artificially synthesized data can accurately know the oil related information and location, automatically generate labeling files, and quickly generate a large amount of sample data. However, due to the limitation of diversity and embedding effect, synthesized data is usually difficult to fit the test set effectively. It is necessary to effectively optimize the sample and background of the synthesized image to integrate the oil into the natural scene picture naturally, so as to improve the quality of the training set. First, the target to be detected is generated, then, edge detection, affine transformation, and other methods are used to better isolate the required markers, and then

bilinear interpolation, salt and pepper noise, and Gaussian blur are used to make target and background integrated.

*2.1. Experimental Setup*

The experiments were carried out in the experimental tank of the key laboratory of marine oil spill identification and damage assessment technology of the State Oceanic Administration. The stainless steel and glass tank (see Figure 1) is 32 m long, 0.8 m wide, and 2 m deep, and the seawater depth is 1.014 m. The main equipment in the experiment includes tantalum wire wave gauge, laser, charge-coupled device (CCD), and ADV. Particle image velocimetry (PIV) is used to collect the flow field changes at each time. The resolution of the CCD camera is 2592 × 2048 pixels with a sampling interval of 0.025 s. The experimental equipment is set as shown in Figure 1. In order to ensure the accuracy of the experimental data, a tantalum wire wavemeter is set outside the camera acquisition area to measure the wave height, and the acquisition frequency is 100 Hz. At the same time, the experiment set the fixed velocity time series acquisition point by ADV, and the sampling frequency is 200 Hz.

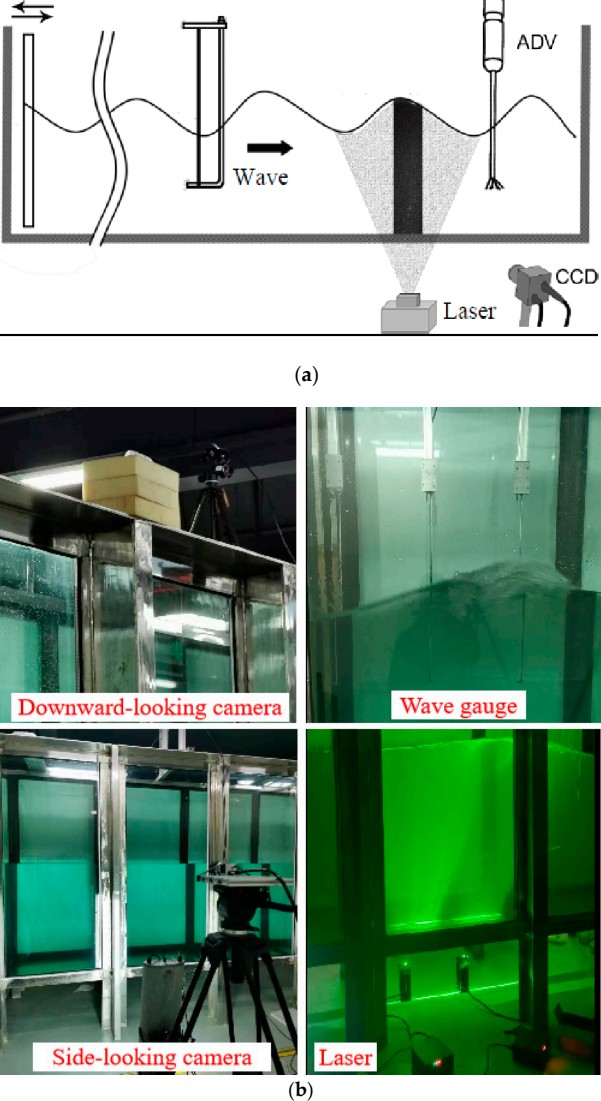

**Figure 1.** Conceptual schematic of the experiment conducted at the laboratory facility. The water tank is shown in (**a**), and the experimental device is shown in (**b**).

*2.2. Generating the Target*

2.2.1. Canny Operator

In order to make the synthetic data set close to the data distribution of the original data set, the Canny edge detection operator is used to calculate the image gradient, eliminate the background of the oil itself, and prevent the background of the oil itself from interfering with the data. The edge of the image usually refers to a local area where the brightness of the area changes significantly in the image. Generally, the gray profile of this area is regarded as a step, that is, a buffer area with a small gray value changes sharply to another gray level. The edge part of the image usually concentrates the information of the huge part of the image, and the edge detection is the measurement of the change in the gray value, and the detection and positioning are performed. Canny edge detection operator is the first commonly used one-dimensional signal, and the complete solution can be obtained by calculus. If the Canny operator is used in a multi-dimensional image, the step edge needs to be obtained through information such as position, direction, and amplitude. Edge detection is usually not processed on the original data. First, the original image is convolved with a symmetrical 2D Gaussian smoothing template to obtain an image slightly blurred than the original image and then differentiated along the direction of the gradient to obtain a simple and effective direction Operator. This series of operations makes the image more stable, and individual pixel noise becomes almost no effect after Gaussian smoothing. The Canny operator contains two sets of $3 \times 3$ matrices, which are convolved with the image, and the horizontal and vertical brightness difference approximate values can be obtained respectively, and then the gradient direction of each pixel of the image can be obtained:

$$G_x = \begin{pmatrix} -1 & 0 & 1 \\ -2 & 0 & 2 \\ -1 & 0 & 1 \end{pmatrix} \cdot A \tag{1}$$

$$G_y = \begin{pmatrix} 1 & 2 & 1 \\ 0 & 0 & 0 \\ -1 & -2 & -1 \end{pmatrix} \cdot A \tag{2}$$

$$G = \sqrt{G_x{}^2 + G_y{}^2} \tag{3}$$

$$\theta = \arctan(\frac{G_x}{G_y}) \tag{4}$$

where $A$ is the image, $G_x$ and $G_y$ are the convolution results of the horizontal and vertical direction operators, $G$ is the final edge amplitude, and the $\theta$ value is the edge direction.

In actual applications, there are some points with larger gradient values that are not edges, so some measures are needed to filter these points. Common detection methods include thresholding, which filters through thresholds. However, how to choose the threshold is also a difficult point. If the threshold is set too high, the edge contour may be easily broken or key information may be missed. If the threshold is set too low, it will cause too much noise, and some redundant information may be used as features. The low threshold and the high threshold are usually estimated based on the signal-to-noise ratio, and the appropriate scale of the operator depends on the number of objects contained in the image. Usually, multiple different scales are used to save the information obtained at different scales, and then the Canny detection operator is represented by different Gaussian standard deviations.

2.2.2. Affine Transformation

After obtaining the oil sample, it is necessary to perform a geometric transformation on the oil. The vector space is linearly transformed and connected to a translation to transform it into another vector space, which is convenient for further processing of the image by manual or machine. In order to simulate the effects of distortion, deformation,

and occlusion in natural scene data, this paper uses an affine transformation to process oil images. Affine transformation is usually achieved by non-singular linear transformation and translation transformation in geometry to realize the mapping between two spaces. It is usually completed by a non-singular linear transformation followed by a translation transformation. Affine change can be understood as the composition of two functions: translation and linear mapping. In a limited dimension, the affine transformation can be obtained by a transformation matrix *A* and a translation vector *b*. Affine transformation formula can be used by:

$$f(x) = Ax + b \tag{5}$$

In a two-dimensional space, *A* can be decomposed into the following four steps:

1. Zoom: Change the scale of the image, enlarge or reduce the image by *S* times, but the aspect ratio of the image is not changed before and after the transformation.
2. Stretching: Stretching is to enlarge or shrink the image along the abscissa or ordinate direction by *t* times.
3. Distortion: Image distortion is the point-to-point mapping of the image. Distortion can be visually understood as the image being projected onto a curved or specularly reflective surface to achieve the effect of deformation.
4. Rotation: Rotation means that the image rotates counterclockwise around the origin by an angle of *θ* degrees.

Then, *A* can be decomposed into:

$$A = A_1 A_2 A_3 A_4 = \begin{pmatrix} s\cos\theta & stu\cos\theta - st\sin\theta \\ s\sin\theta & stu\sin\theta + st\cos\theta \end{pmatrix}, \ 0 \le \theta \le 2\pi \tag{6}$$

Transformation of each pixel:

$$\begin{pmatrix} x' \\ y' \end{pmatrix} = \begin{pmatrix} s\cos\theta & stu\cos\theta - st\sin\theta \\ s\sin\theta & stu\sin\theta + st\cos\theta \end{pmatrix} \begin{pmatrix} x \\ y \end{pmatrix}, \ 0 \le \theta \le 2\pi \tag{7}$$

*2.3. Target and Background Fusion*

2.3.1. Bilinear Interpolation

After obtaining the oil image sample, we will place the oil in a suitable location. During the rendering process, the oil can be the same color as the background, but the color value must be increased or decreased. In the fusion process, a pixel in the oil map is often mapped to a position between several pixels in the background image, so a new pixel needs to be obtained through interpolation. Interpolation methods are usually divided into two categories: forward mapping and backward mapping.

Forward mapping traverses each pixel in the image and calculates the position of the point in the new image according to the mapping function. The new position can directly select the corresponding pixel value in the original image or an approximate value of the original pixel value. The forward mapping method may result in some holes in the new image, that is, all pixels in the original image cannot be mapped to the location of this hole. In addition, there may also be two different pixels mapped to the same location, which will result in the loss of some pixel information. A typical approach in forwarding mapping is nearest-neighbor interpolation.

Backward mapping is the opposite of forward mapping. It is to reversely find the mapping transformation from the new image to the original image, and then traverse each pixel in the new image to calculate its corresponding position in the original image. This position may not be an integer either. The simple way is to round it directly. In order to achieve better results, you can also take the average of the points around the synthesis. The typical approach in forward mapping is a bilinear interpolation.

In order to make the background blend into the background image more naturally, this paper performs bilinear interpolation on these pixels. Bilinear interpolation is the expansion of one-dimensional linear interpolation in two dimensions. It is completed by

performing linear interpolation in two dimensions respectively. Compared with the simple and crude nearest-neighbor interpolation and linear interpolation, the effect of bilinear interpolation is usually better, and the amount of calculation and ease of understanding is better than higher-order interpolation methods such as bicubic interpolation and cubic spline interpolation.

### 2.3.2. Salt and Pepper Noise

Image noise refers to the redundant or interfering information in the image, and noise will affect the quality of the image. However, natural scene images often have more or less noise. Noise may come from many aspects, such as flicker and noise caused by electronic components, particles and noise caused by junction transistors, road heat noise and so on. There is always noise in the image. In order to make the quality of the synthesized data close to the quality of the original picture, it is necessary to imitate the noise in the original picture and add random noise to the image.

Salt and pepper noise is the black and white bright and dark spots caused by the image sensor, transmission channel, and decoding process. Salt and pepper noise refers to two kinds of noise, salt noise, and pepper noise. Salt noise is white and pepper noise is black noise. The former is high-gray noise, and the latter is low-gray noise. Generally, two kinds of noise will appear at the same time, and the image will appear as black and white noise. This article controls the amount of noise and randomly distributes black and white noise in each channel of the image. SNR is a random value with a value range of (0.9, 1).

### 2.3.3. Gaussian Blur

In order to make the synthetic data distribution closer to the original data distribution, this paper adds fuzzy to imitate some disturbances that often appear in natural scene data. The blurring process resets the pixels in the image and performs a weighted average according to the gray values of the pixels to be filtered and the surrounding pixels. Therefore, how to choose the weight is a problem that needs to be studied. If the simple average is used, it is obviously too average. The points of the image are continuous, and the closer the distance is, the higher the correlation, so the weight value should also change with the change in the distance. The original pixels have the largest Gaussian distribution, in this way the edge results are better preserved. In this paper, Gaussian blur is used to reduce image noise and reduce the level of detail, thereby enhancing the effect of the image at different scales. In geometry, Gaussian blur is obtained by convolution of an image matrix and a density function with a normal distribution, which is equivalent to low-pass filtering of the image. It makes the effect of image out-of-focus imaging more obvious. The corresponding Gaussian calculation formula is shown in the following.

$$G(r) \approx \frac{1}{\sqrt{2\pi\sigma^2}^N} e^{\frac{-r^2}{2\sigma^2}} \tag{8}$$

where $\sigma$ is the standard deviation of the normal distribution, the larger the value of $\sigma$, the more blurred the image. $r$ is the blur radius, that is, the distance from the template element to the center of the template. This paper uses the standard normal distribution of $\sigma = 1$, and $r$ is a random value in the range of (2, 8).

### 2.4. Data Preprocessing

In deep learning, due to the complexity of the model and too many parameters, the error in the noise may also be fitted during fitting, resulting in excessive variance. In order to reduce the overfitting of the model, data preprocessing is usually required. Data preprocessing is usually to expand the data, increase the amount of data through strategies such as color transformation and random tailoring, and normalize the data.

In order to alleviate the over-fitting phenomenon of deep neural networks, random tailoring is often used to expand the data set. Random cropping refers to randomly cropping a part of the image, and doing horizontal mapping. Then it is the input of the

network for training. When randomly cropping, the main body part should be cut out and included in the training set, which is equivalent to enriching the data set in the dimension of position. At the same time, it adds the opposite direction graph to make the training set more diversified. Random cropping can generate new pictures to expand the data set. Although the human eyes can easily judge that the original picture is the same picture after random cropping, for the computer that treats the pictures as an array, the picture is very different from the original picture. As shown in Figure 2, the principle of clipping is not to destroy the main structure of the original image, and not to cut the edges of the object too much, otherwise it will lose the information of the object. In addition, when cropping, it is necessary to control the area ratio of the cropped area, the aspect ratio of the area, the target threshold, the number of random crops for each picture, etc., to prevent the cropped image from being distorted.

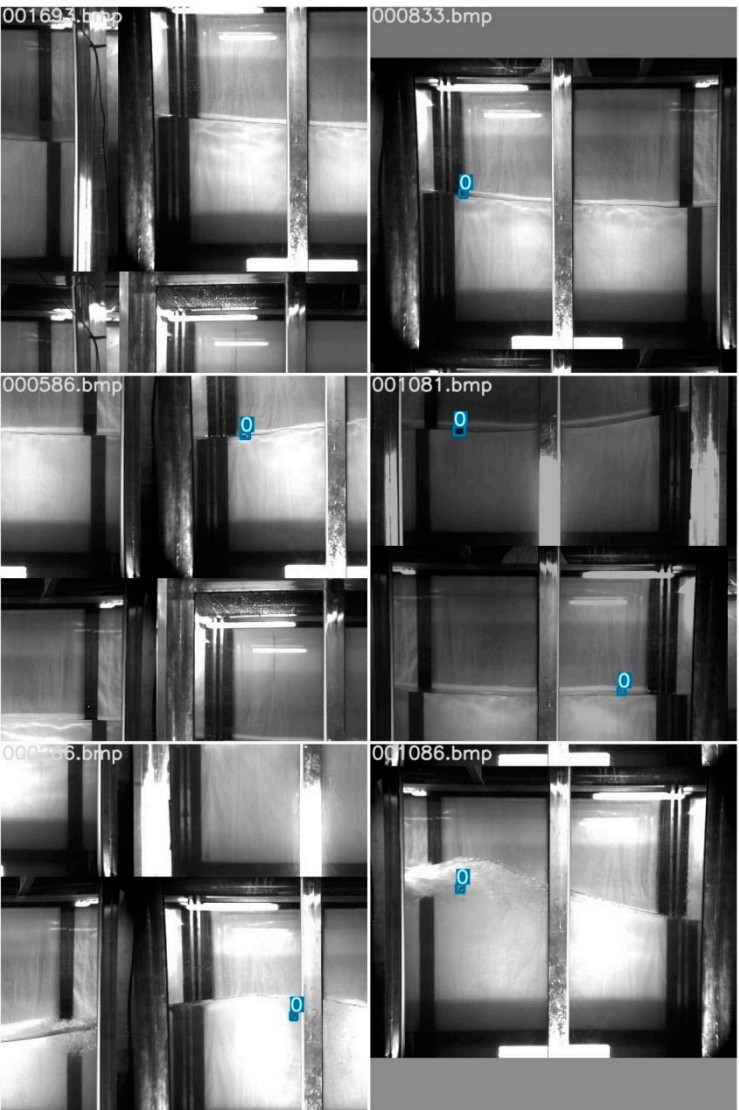

**Figure 2.** The performance of image enhancement during training.

### 2.5. Sample Design for Training the Classifier

The method based on multi-strategy fusion is used to improve the accuracy of small object detection. First, Feature Pyramid Networks adds upsampling and pixel values so that low-level texture features of the image extracted by ResNet are combined with high-level semantic features. It reduces the serious loss of location information of small objects after multiple sampling. Second, RoIAlign cancels the quantization operation to

make the local feature mapping more accurate, which improves the accuracy of feature mapping of small target objects. Finally, the small target objects with high confidence are detected. It integrates the difficulty feedback of entropy as a sample into the loss, increases the contribution of the difficult sample such as a small object.

After obtaining the feature map of fixed size, the feature map is connected with the fully connected layer. It is usually divided into two branches. One branch classifies the feature map to obtain the category of the region, and the other branch performs border regression (get the discrete coordinate value of the border). Finally, each region has two output values: the category it belongs to and the corresponding confidence, and the fine-tuned coordinates. In the multi-classification problem, the Softmax classifier is used in this paper.

### 2.5.1. Softmax

The Softmax function, also called normalized exponential function, is a generalized multinomial logistic regression model, which extends logistic regression to multi-class recognition and is used for multi-classification problems. In multi-class classification, Softmax outputs a floating-point number representing the probability for each class, and the sum of the probabilities of all classes is 1. This helps the model's training process to converge faster. The calculation of Softmax is shown in Equation (9).

$$p_j = \frac{e^{a_j}}{\sum\limits_{k=1}^{N} e^{a_k}} \tag{9}$$

where $p$ is the output vector, and it contains the probability value of the region belonging to each category, and $p_j$ represents the probability corresponding to the $j$th category. $N$ is the total number of categories. $a$ is the output vector of the fully connected layer. The output of each neuron is firstly subjected to exponential operation, and the probability value of each category is obtained by using the exponential sum of the corresponding value of the category in all categories.

### 2.5.2. Accuracy Assessment

In order to verify the performance of the model, four indicators including the precision, recall, *mAP*, and detection speed were adopted for the evaluation in this study. When the *IOU* (Intersection Over Union) $\geq 0.5$, it is a true case. When the *IOU* < 0.5, it is a false positive case. When the *IOU* = 0, it is a false negative case. The calculations of the *IOU*, precision, recall, and *mAP* were shown in Equations (10)–(13). Here, the *mAP* is the average value of the *AP* (Average Precision) when the oil is detected; and the higher the value is, the better the detection result of liquid oil.

$$IOU(R, R') = \frac{|R \cap R'|}{|R \cup R'|} \tag{10}$$

$$precision = \frac{TP}{TP + FP} \times 100\% \tag{11}$$

$$recall = \frac{TP}{TP + FN} \times 100\% \tag{12}$$

$$mAP = \frac{\sum_{c=1}^{C} AP(c)}{C} \tag{13}$$

where $R$ is the detected area of the object bounding box; $R'$ is the actual area of the object bounding box; *TP*, *FP*, and *FN* are the numbers of true positive cases, false positive cases and false negative cases, respectively; and $C$ is the number of detection categories. Since only oil was detected in this study, $C = 1$ was used in this study.

This paper uses the feature pyramid-based region proposal network as the method for extracting candidate regions. The image size is 416 pixel × 416 pixel. Batch_size is 1, and the number of RoI in each batch is 256, then the ratio of foreground and background is 1:1. The learning rate is 0.0025, and the learning rate is reduced in stages. The maximum number of iterations is 60,000, and the weight decay is 0.0001.

## 3. Results

### 3.1. Generating Labeled File

This research has used mosaic data enhancement, random flipping, and Gaussian noise fuzzy data enhancement, as well as cosine annealing learning rate, and label smoothing to improve the effect of the image. When the object to be detected is fused with the background, we can clearly know the category and location of the target object, so we can automatically generate the labeled image file. This article uses MSCOCO json file format for labeling, including five basic types: info, image, license, annotation, and categories.

The algorithm of ImageLabel 1 json:
{
"info":info,
"licenses": [license],
"images":[image].
"annotations":[annotation],
"categories":[categories],
}

Info is the information description of the data set.

The data analyzed in this article all use the area of Figure 3, namely the upper left point $x_1$ (566,961), and the lower right point $x_2$ (1182,1702). The reason for choosing this area is that the single glass area excludes the influence of the objects in the blue box. Thus, this paper records the total number of pixels considered as liquid oil in each liquid oil area. The results of the study found that the selection of the region can eliminate some interference to a certain extent, and the selection of the segmentation threshold determines the condition of the segmented oil and liquid regions.

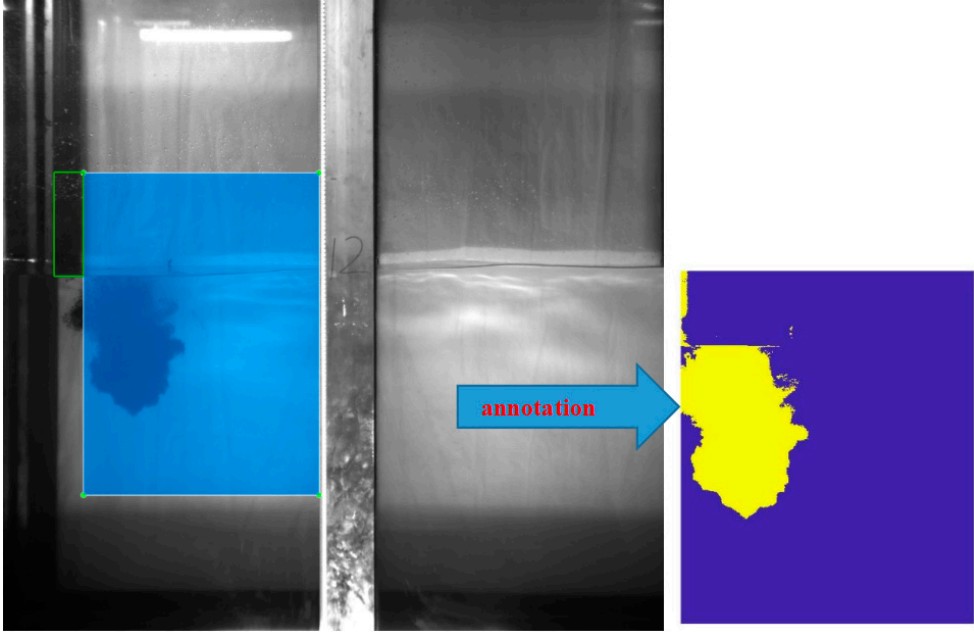

**Figure 3.** Sunken and submerged oil in the observation area.

### 3.1.1. Comparison between 6 Experiments

Experiment (1): Basic experiment (RoIPool). As the benchmark value of a series of experiments, the basic experiment not only test the possibility of oil detection, but also provid a reference standard for experiments (2) to (6). The subsequent experiments also optimize the basic experiment to a certain extent. This article uses Faster RCNN framework + Resnet-50 basic network as the basic experiment. Faster RCNN is a classic two-stage target detection framework. The best method for the MSCOCO data set is to improve Faster RCNN. Resnet-50 is a 50-layer residual network with high computational accuracy and low computational complexity, and it is an ideal training model for residual networks. In addition, the basic experiment does not use a feature pyramid. It uses a region proposal network as a method of extracting candidate regions, and it uses RoIPool for candidate region mapping.

Experiment (2): RoIAlign. On the basis of experiment (1), RoIPool is changed to RoIAlign, and other conditions are unchanged. Then, we can compare and observe the experimental effect of RoIAlign.

Experiment (3): FPN. In order to verify the influence of feature fusion on the oil detection effect, on the basis of experiment (1), the single-layer feature prediction is changed to a feature pyramid, and the feature pyramid takes 2–6 layers, and the anchor selects three ratios of {1:2, 1:1, 2:1} and five dimensions of {32, 64, 128, 256, 512}, and other conditions remain unchanged. Then we can observe the experimental results.

Experiment (4): Retinanet. In order to verify the accuracy of the two-stage detection method represented by Faster RCNN + Resnet-50 + FPN, this paper selects the Retinanet as a comparative experiment. Retinanet has the highest accuracy among one-stage methods. Retinanet is a new contribution of the Facebook AI team in the field of target detection. It is a combined application of FPN and FCN networks. It has good experimental results in MSCOCO and other data sets.

Experiment (5): FPN + RoIAlign. The experiment combines the feature fusion FPN with the improved feature mapping RoIAlign to observe the effect of oil detection after the fusion of multiple methods.

Experiment (6): FPN + RoIAlign + synthetic data. On the basis of experiment (5), synthetic data is added to observe the effect of oil detection when adding different amounts of data, so as to obtain the effect of synthetic data on oil detection.

### 3.1.2. Oil Detection

The detection results of the model in the original data set are shown in Table 1. The framework used in experiment 1 (baseline) has an *AP* (accuracy percent) of 0.358 on the authoritative data set of MSCOCO. Although the data set of this subject has fewer categories than the object categories in MSCOCO, its data concentrate small objects accounted for more than 90%, and MSCOCO small objects accounted for only 31.4%. So, the accuracy of the experiment has not been greatly improved. Comparing experiments 1, 2, and 1, 3, it can be seen that the use of feature fusion and improved feature mapping can increase *AP* by 62.29% and 67.60%, respectively, indicating that these two improved methods are particularly effective for object detection. Object detection has been greatly improved. Results of experiments 4 and 5 indicate that Faster RCNN + Resnet methods are superior to other methods in basic feature extraction. FPN + RoIAlign + synthetic data in experiment (6) can increase *AP* by 0.73%, compared with experiment (5). As the amount of synthetic data increases, the accuracy of oil detection continues to increase. It indicates that the synthetic data proposed in this paper can effectively improve the results of oil detection.

For oils that are not labeled in the training set, the model test results are shown in Table 2. It can be seen from the test results that the model has a good generalization ability even for oils without labeling in the training set. For these oils, the use of synthetic data greatly improves the detection effect. As the number of synthetic data increases, the *AP* gradually increases. In addition, *AP* of FPN + RoIAlign + synthetic data method for labeled oil detection is 18.56% higher than that of oil detection without labeling.

**Table 1.** Comparison of the labeled oil detection results in the training set.

| No. | Methods | Synthetic Data | *AP* | *AP50* | *AP75* |
|---|---|---|---|---|---|
| 1 | RoIPool | 0 | 0.358 | 0.856 | 0.200 |
| 2 | RoIAlign | 0 | 0.581 | 0.918 | 0.680 |
| 3 | FPN | 0 | 0.600 | 0.951 | 0.722 |
| 4 | Retinanet | 0 | 0.6103 | 0.944 | 0.759 |
| 5 | FPN + RoIAlign | 0 | 0.6265 | 0.952 | 0.774 |
| 6 | FPN + RoIAlign + synthetic data | 200 | 0.6311 | 0.953 | 0.777 |
| 7 | FPN + RoIAlign + synthetic data | 600 | 0.6322 | 0.966 | 0.788 |
| 8 | FPN + RoIAlign + synthetic data | 1838 | 0.6388 | 0.977 | 0.789 |

**Table 2.** Comparison of oil detection without labeling in the training set.

| No. | Methods | Synthetic Data | *AP* | *AP50* | *AP75* |
|---|---|---|---|---|---|
| 1 | RoIPool | 0 | 0.298 | 0.656 | 0.155 |
| 2 | RoIAlign | 0 | 0.481 | 0.718 | 0.480 |
| 3 | FPN | 0 | 0.470 | 0.751 | 0.522 |
| 4 | Retinanet | 0 | 0.3103 | 0.544 | 0.559 |
| 5 | FPN + RoIAlign | 0 | 0.3265 | 0.752 | 0.574 |
| 6 | FPN + RoIAlign + synthetic data | 200 | 0.5311 | 0.753 | 0.577 |
| 7 | FPN + RoIAlign + synthetic data | 600 | 0.5322 | 0.766 | 0.688 |
| 8 | FPN + RoIAlign + synthetic data | 1838 | 0.5388 | 0.907 | 0.689 |

With the continuous increase in the amount of synthetic data, the recognition effect of the two test sets have been continuously improved. In accordance with the distribution of the original data, the training effect of the synthetic data is close to that of the original data training. Even if the oil category does not appear in the training set, it can be effectively identified through the synthetic data, which solves the problem that the current oil pictures are difficult to obtain. Adding 1838 synthesized data in the training set, the average accuracy of the oil detection is increased by 79.72%, which solves the current defects of the difficulty of obtaining oil-moving images and the high cost of image labeling.

### 3.2. Detection of Sunken and Submerged Oil

3.2.1. Breaking Wave with 15 cm Wave Height

Under the action of a 15 cm breaking wave, the liquid oil is hit and sunk to form sunken and submerged oil. Set the initial time when the breaking wave hits the oil, which is shown in Figure 4. When $t$ = 6.92 s, the oil is brought to the position of Figure 5 by the wave, that is, the initial time of diving. When $t$ = 7.63 s, the oil was migrated by a 15 cm breaking wave in Figure 6. Then, the oil remains below the water surface until it surfaces again under the action of buoyancy to form suspension (Figure 7). During this period, the oil stays under the water for nearly 3.0 s. Meanwhile, the oil mass can be observed by the downward-looking camera again. Under the action of a 15 cm breaking wave, the oil mass reaches the deepest position of 180.62 mm, namely 0.181 m. The horizontal displacement is 1.25 m.

3.2.2. Breaking Wave with 25 cm Wave Height

Under the action of a 25 cm breaking wave, the liquid oil is hit and sunk to form sunken and submerged oil. Set the initial time when the breaking wave hits the oil, which is shown in Figure 8. When $t$ = 6.76 s, the oil is brought to the position of Figure 9 by the wave, that is, the initial time of diving. When $t$ =7.16 s, the oil was migrated by a 25 cm breaking wave in Figure 10. Then, the oil remains below the water surface until it surfaces again under the action of buoyancy to form suspension (Figure 11). During this period, the oil stays under the water for nearly 3.5 s. Meanwhile, the oil can be observed by the downward-looking camera again. Under the action of a 25 cm breaking wave, the oil

reaches the deepest position of 205.00 mm, namely 0.205 m. The horizontal displacement is 1.90 m.

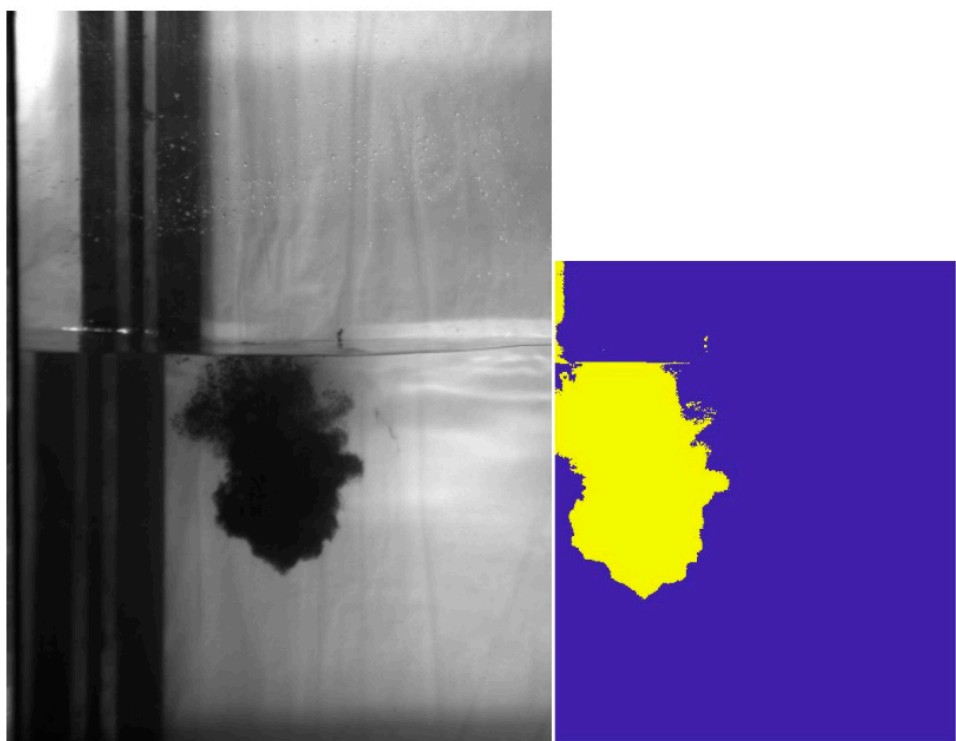

**Figure 4.** Spilled oil in the observation area without breaking wave.

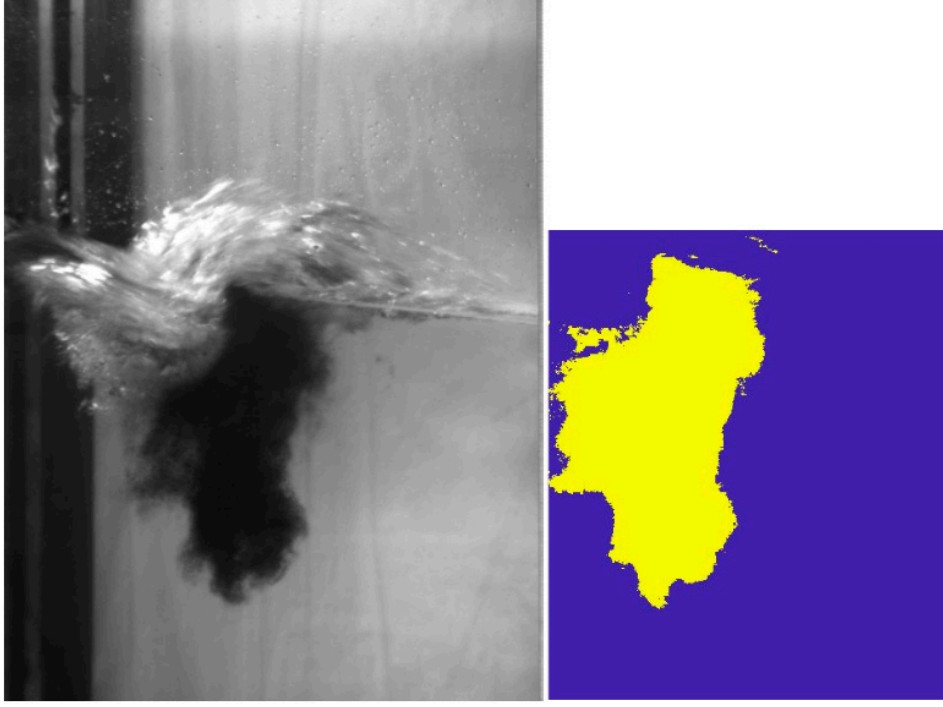

**Figure 5.** Spilled oil in the observation area was hit by 15 cm breaking wave.

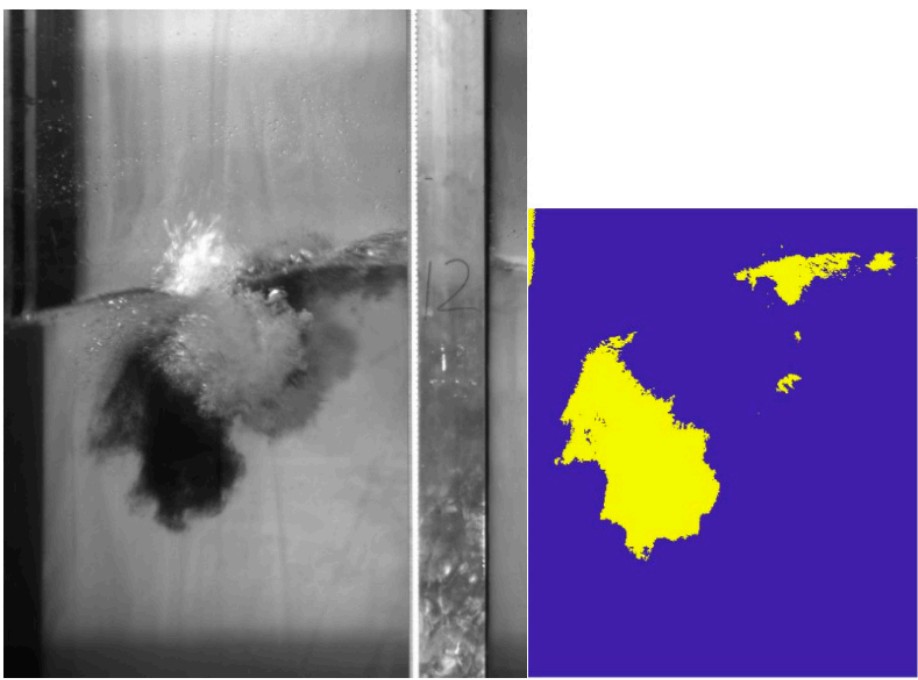

**Figure 6.** Spilled oil in the observation area was migrated by 15 cm breaking wave.

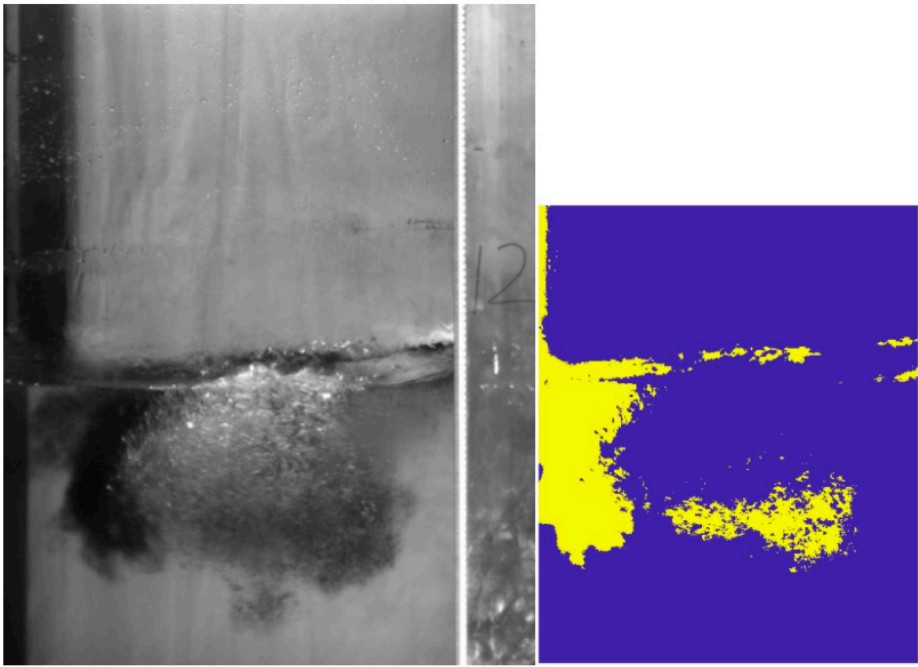

**Figure 7.** Spilled oil in the deepest position of the observation area under a 15 cm breaking wave.

### 3.2.3. Breaking Wave with 35 cm Wave Height

Under the action of 35 cm breaking wave, the liquid oil is hit and sunk to form sunken and submerged oil. Set the initial time when the breaking wave hits the oil, which is shown in Figure 12. When $t$ = 1.60 s, the oil is brought to the position of Figure 13 by the wave, that is, the initial time of diving. When $t$ = 2.40 s, the oil was migrated by 35 cm breaking wave in Figure 14. Then, the oil remains below the water surface until it surfaces again under the action of buoyancy to form suspension (Figure 15). During this period, the oil stays under the water for nearly 4.2 s. Meanwhile, the oil can be observed by the downward-looking

camera again. Under the action of 35 cm breaking wave, the oil reaches the deepest position of 250.00 mm, namely 0.25 m. The horizontal displacement is over 2.00 m.

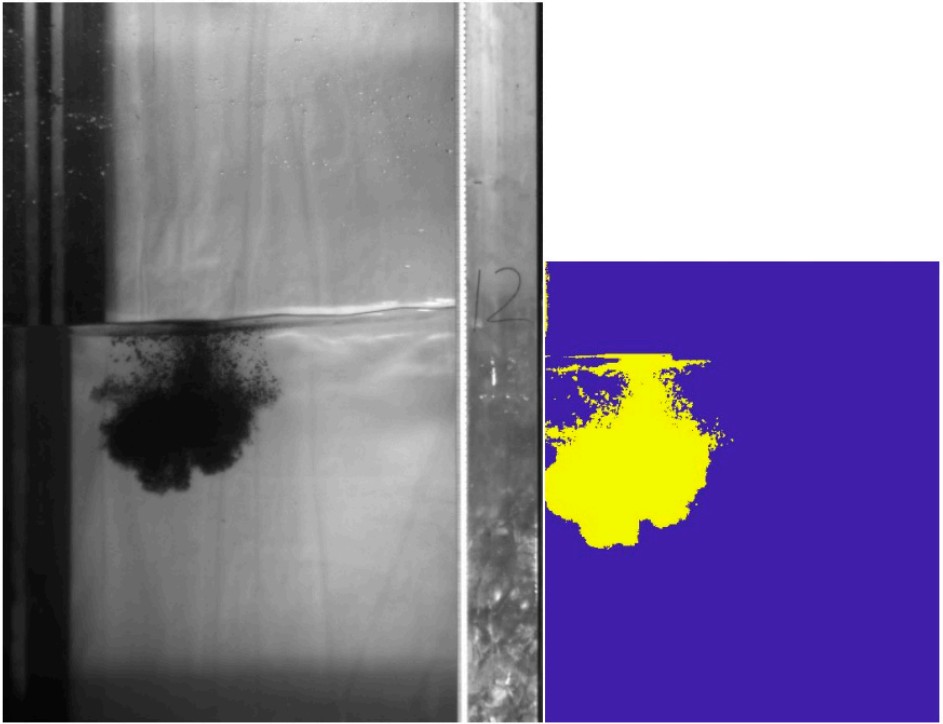

**Figure 8.** Spilled oil in the observation area without breaking wave.

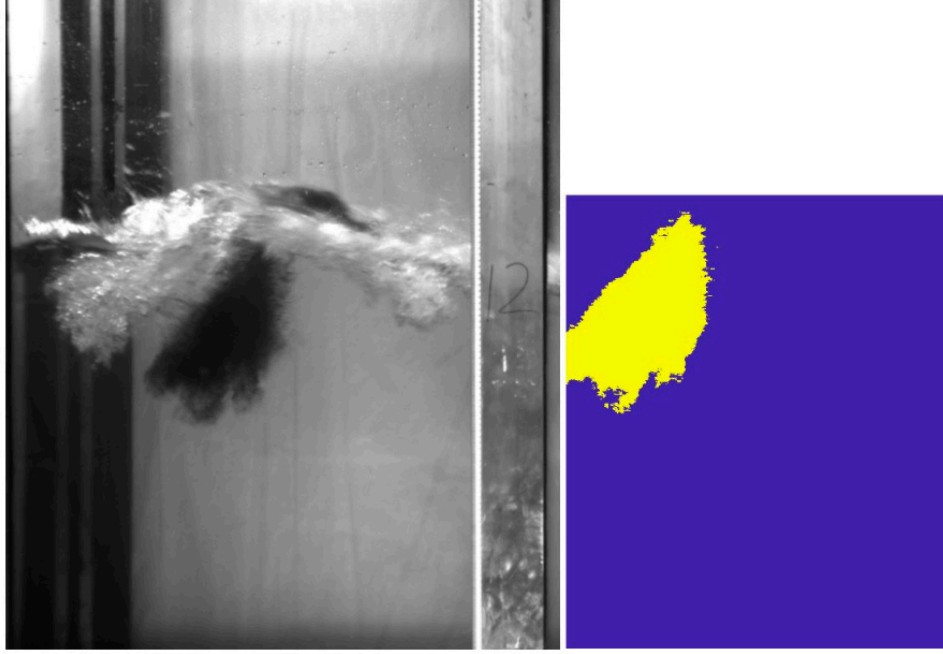

**Figure 9.** Spilled oil in the observation area was hit by 25 cm breaking wave.

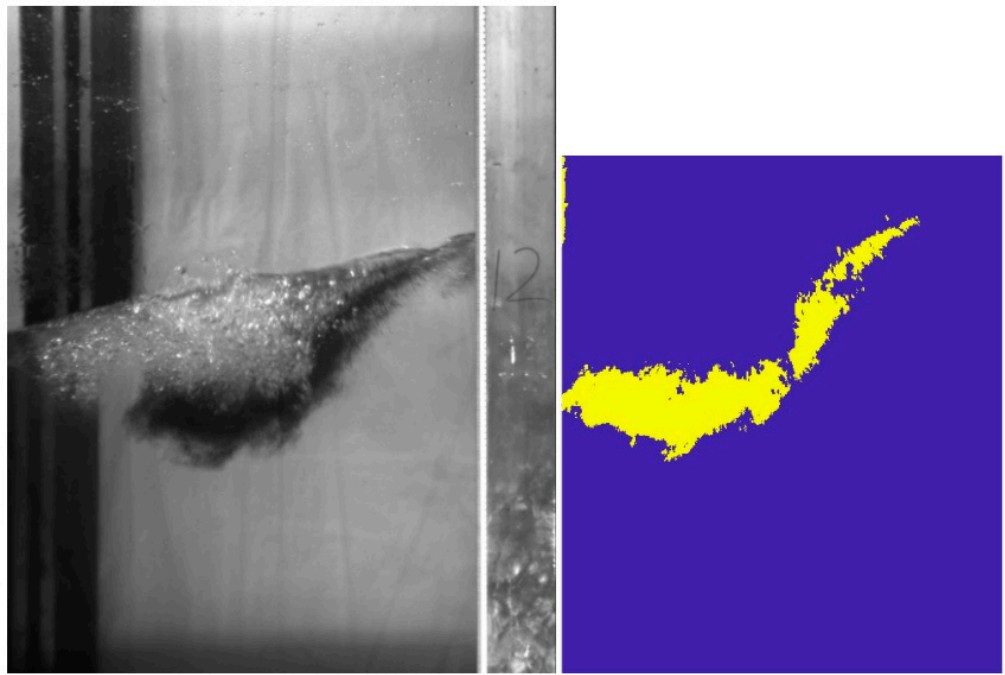

**Figure 10.** Spilled oil in the observation area was migrated by 25 cm breaking wave.

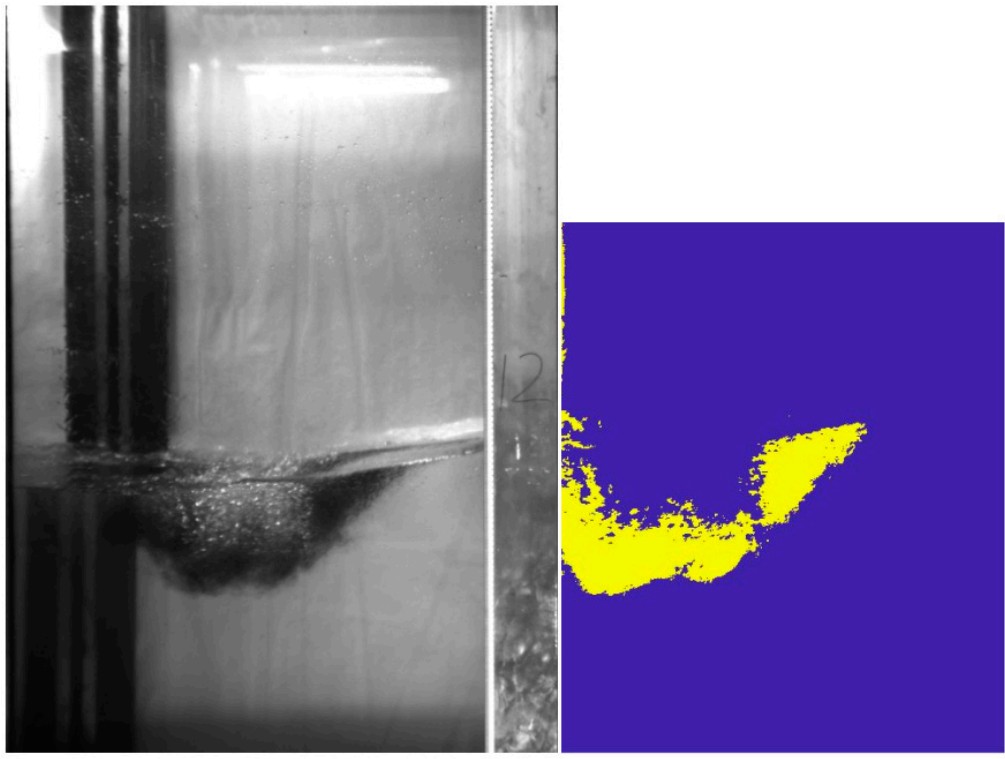

**Figure 11.** Spilled oil in the deepest position of observation area under 25 cm breaking wave.

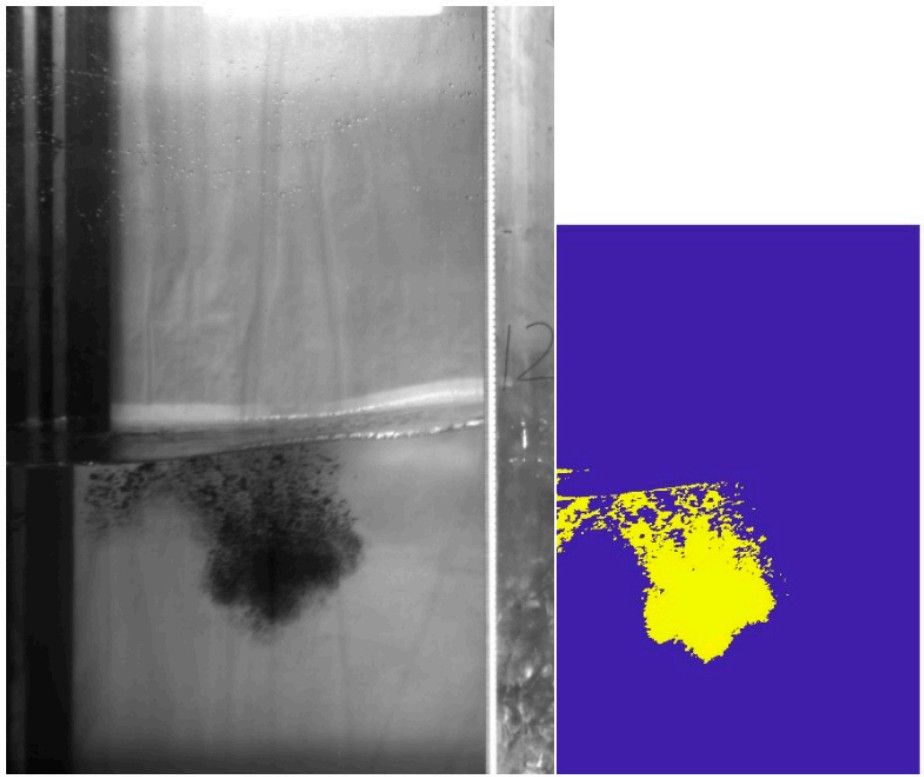

**Figure 12.** Spilled oil in the observation area without breaking wave.

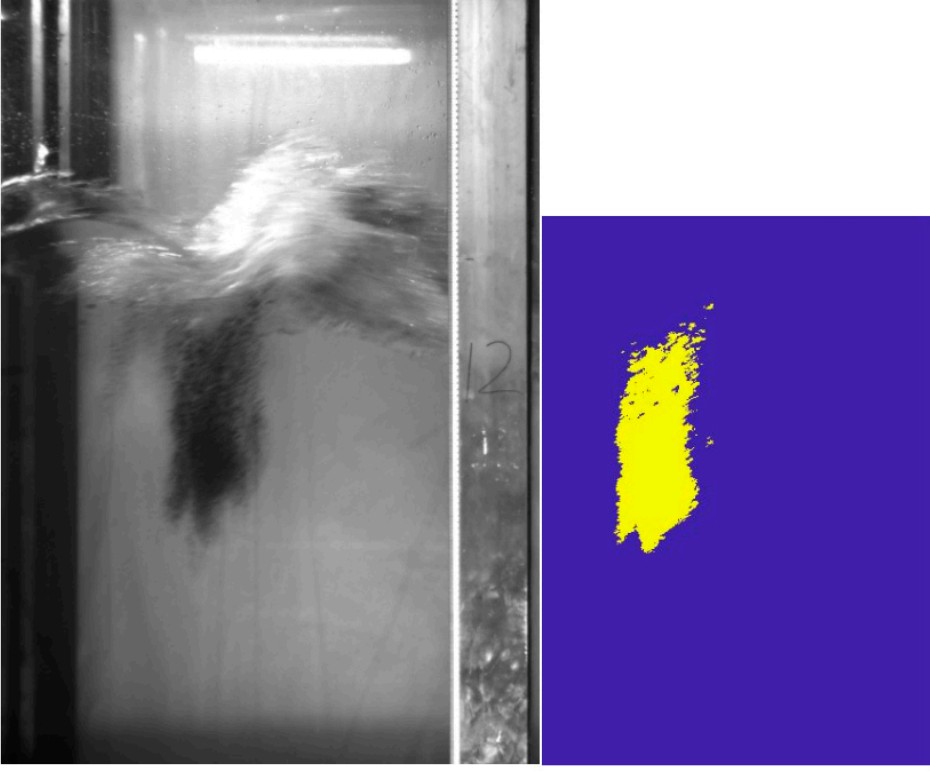

**Figure 13.** Spilled oil in the observation area was hit by 35 cm breaking wave.

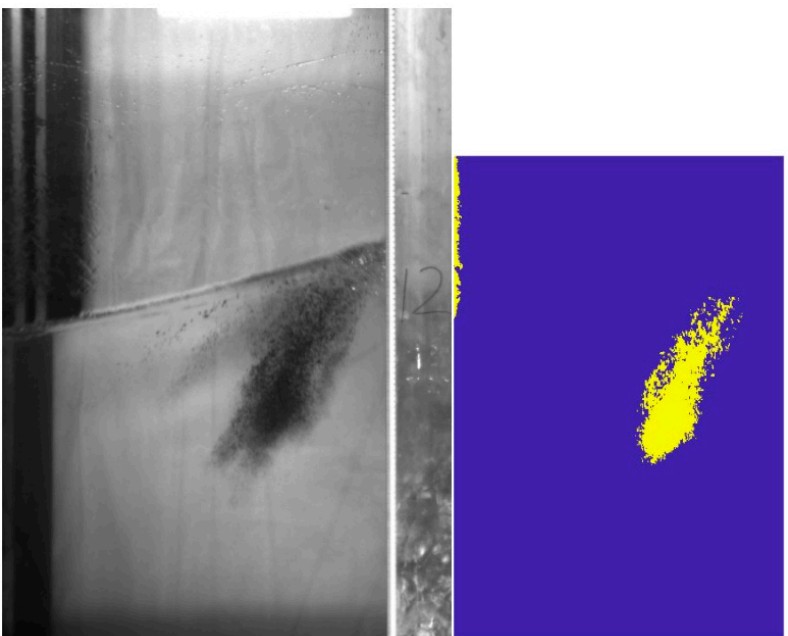

**Figure 14.** Spilled oil in the observation area was migrated by 35 cm breaking wave.

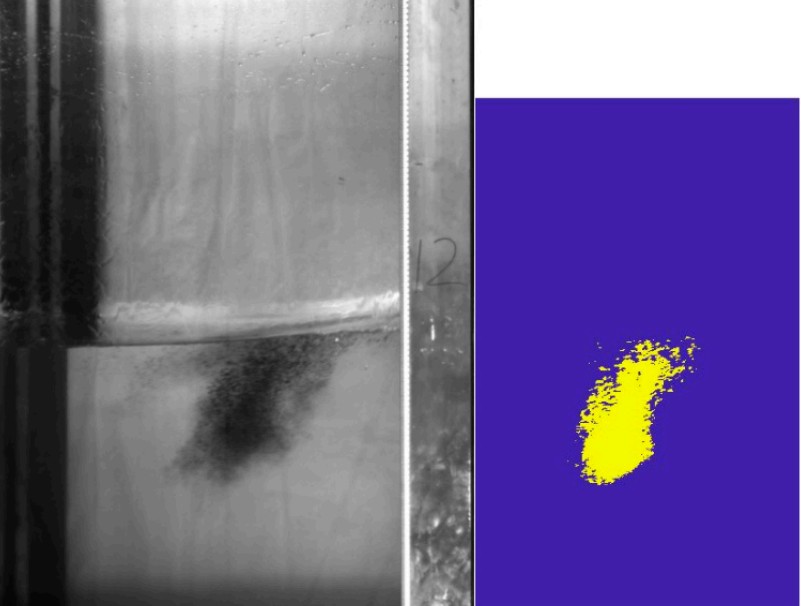

**Figure 15.** Spilled oil in the deepest position of observation area under 35 cm breaking wave.

## 4. Discussion

After the position of the breaking point is obtained, the position of the breaking wave is determined by comparing it with the wave position calculated by the up zero-crossing point, and the proportion of the number of breaking waves in the total wave. That is, the occurrence rate of wave breaking. When the significant wave height is large, the increase of wave energy has a very significant effect on the corresponding increase of wave breaking rate, and the relationship between the two is square-law [58]. When the wave height is small, the wave structure is stable and the wave surface is relatively smooth. After increasing the wave height, the wave instability increases, and the waveform begins to become irregular, and breaking occurs from time to time. When the wave height continues to increase, the wave breaking frequency is very high, and the wave surface breaks and mixes violently with the air. The turbulent energy mainly comes from wave breaking, and its intensity

represents the turbulent mixing intensity during wave breaking. In the experiment, the turbulent energy dissipation rate is used to evaluate the change of turbulence during wave breaking. The calculation of turbulent energy dissipation rate depends on the PIV flow field. Through the analysis of the convection field, it is found that the wave will produce a forward average flow in the forward process [59]. The influence of the average flow is eliminated in the velocity time series of a single point, and then it is band-pass filtered to remove the periodic velocity to obtain the time series of fluctuating velocity. According to the PIV measurement results, the spatial distribution of turbulent energy dissipation rate is obtained by calculating the wave motion process at different times, and then the variation of turbulence in the process of wave propagation is analyzed.

Figure 16 shows that when the wave breaks, the turbulence is mainly concentrated on the wave crest surface, and the water–air interface changes irregularly. When the wave surface is discontinuous, the wave begins to break, and part of the water column on the wave surface separates from the wave surface and enters the air. After a period of time, the water column falls, and it entrains some bubbles into the water, and there is a layer of water air turbulence mixing zone on the water–air interface. In this region, the turbulent kinetic energy increases sharply and the turbulent activity increases. With the diffusion of turbulence, the turbulence influence depth increases, and the mixing increases. It is observed that the area covered by the white crown enhances with the increase of significant wave height. The bubble content in the water suddenly grows when the wave breaks, and the motion trajectory is relatively random. Then, due to the influence of buoyancy, the bubble floats up to the water–air interface again and escapes into the air [60]. The momentum and energy exchange between the bubble and the water body occurs during the movement process, and the turbulent movement is accompanied by the bubble, resulting in the enhancement of the overall turbulent mixing. At $t$ = T/4, the wave surface begins to change irregularly, and the wave steepness begins to increase, resulting in obvious breaking and mixing. In this process, the turbulent energy dissipation rate will increase to a certain extent. With some air entering the water, a large number of bubbles will be generated near the water–air contact surface to strengthen the mixing of the surface. When the wave height reaches the maximum, strong turbulence is generated near the wave surface after breaking, and the turbulence is gradually dispersed in the whole influence depth. The turbulent energy dissipation is always large at the interface, indicating that there is a strong turbulent disturbance here.

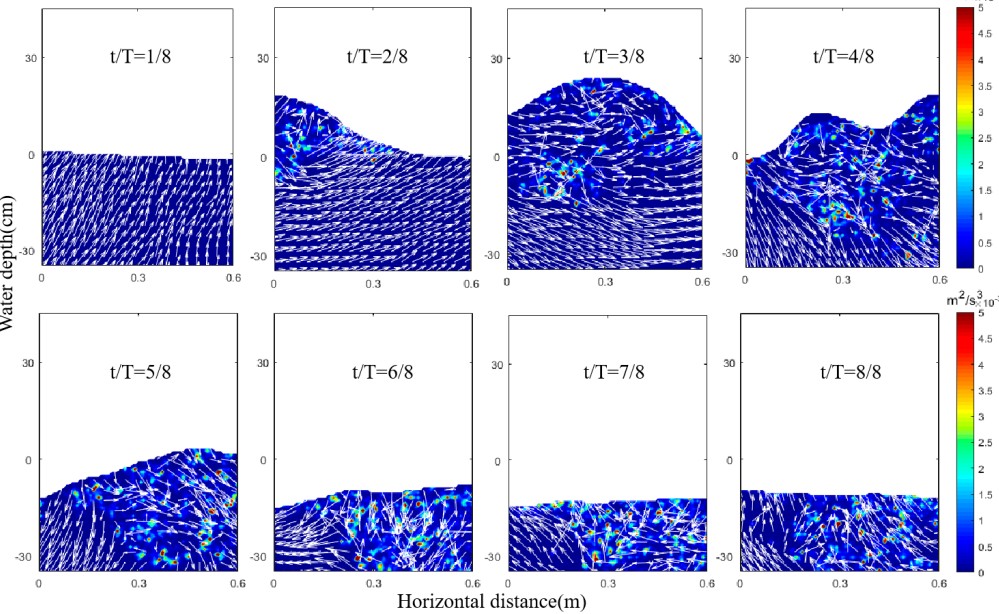

**Figure 16.** Spatial distribution of turbulent dissipation rate for 15 cm breaking wave.

After about 1.0 s, the turbulence intensity gradually decreases due to viscous dissipation and then disappears. This result is similar to the result of Deane et al. [61], who measured the turbulence noise in the wave breaking process (about 1.0 s) in the laboratory. In the actual ocean, this process is reflected in the evolution of the ocean's white crown. Ocean mixing is mainly realized by the disturbance of motion forms such as waves, turbulence, and vortex. When the sea surface waves only fluctuate regularly, the contact surface between the sea surface and the atmosphere is very smooth and complete. At this time, the boundary between the ocean and the atmosphere is very clear. Due to the differences in the physical properties of seawater and air, the sea surface will hinder the energy and material exchange between the ocean and the atmosphere. At this time, the relationship between the two is relatively weak. Only when the wave breaks, the sea surface water and the atmosphere form a mixed layer. As the connecting link between the ocean and the atmosphere, it is mixed with a large number of bubbles and seawater. The contact area between the gas and seawater increases, and the sea air exchange capacity is greatly strengthened. At this time, part of the wave-breaking energy generates turbulence and vortex in the upper mixed layer, and the energy is transmitted to the ocean depth. On the ocean surface, the white crown coverage increases, and the sea surface roughness increases [62]. By calculating the time series of the middle section of the turbulent energy dissipation rate (Figures 17 and 18), it is found that under the medium significant wave height, the occurrence time of the breaking wave has a certain regular period, and the turbulent mixing is mainly concentrated in the breaking wave period. In the nonbreaking wave period, there is little difference between the change of the turbulent flow and the background turbulent field at the bottom of the flume. Comparing the time series of the turbulent energy dissipation rate of different groups, with the rapid increase of the number of breaking waves, the larger the area with a large turbulent energy dissipation rate accounts for the whole wave image, the more important turbulence plays in the exchange of momentum and energy between water and air.

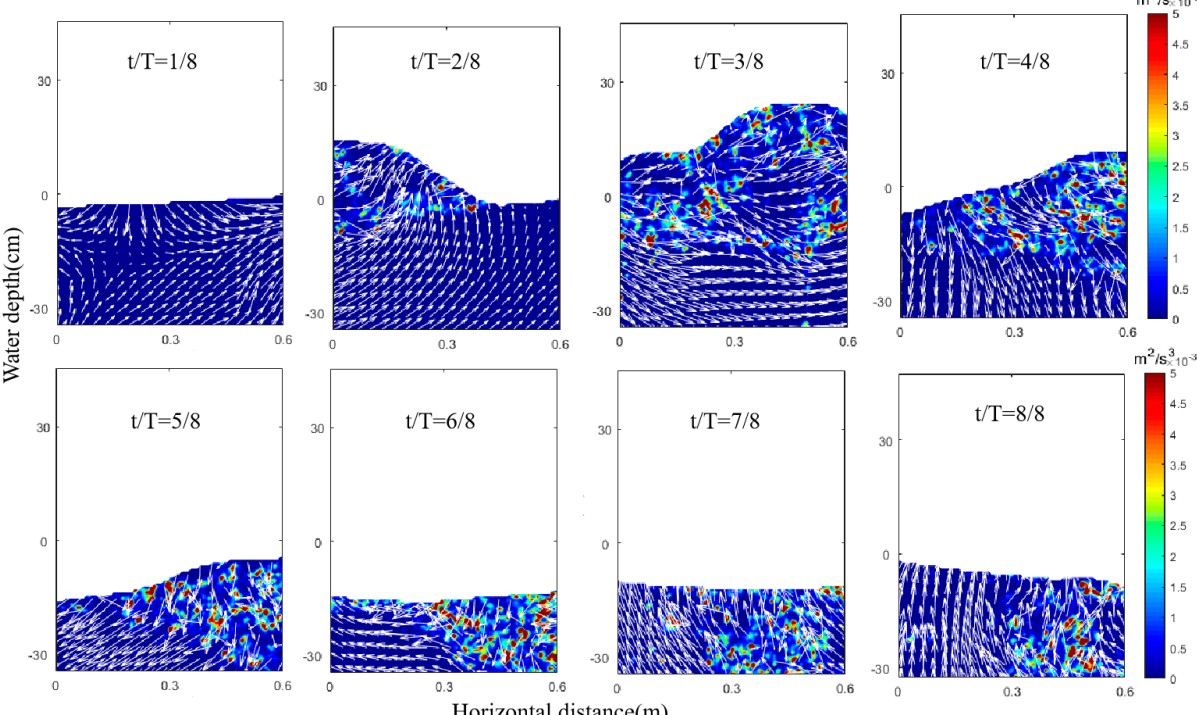

**Figure 17.** Spatial distribution of turbulent dissipation rate for 25 cm breaking wave.

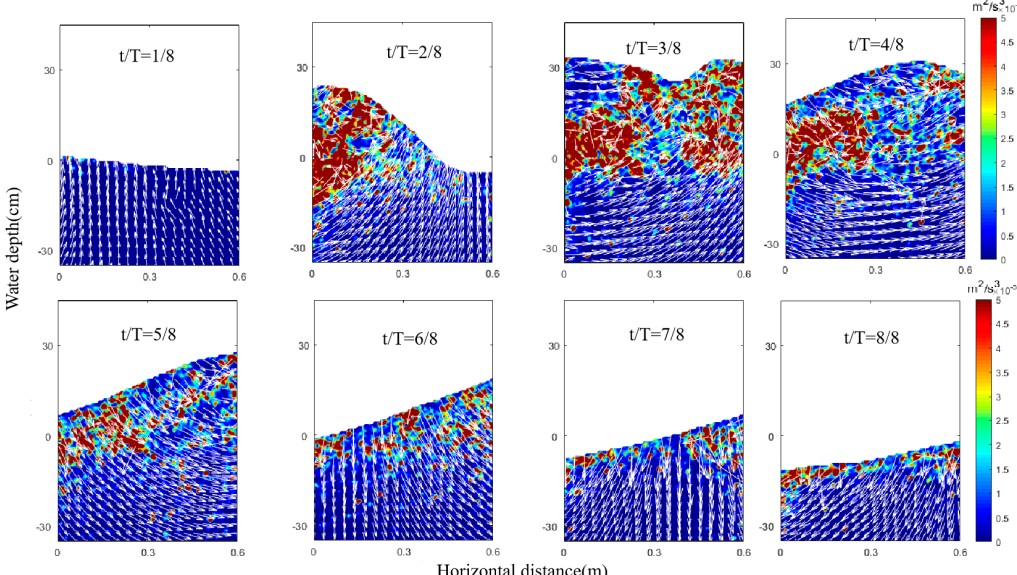

**Figure 18.** Spatial distribution of turbulent dissipation rate for 35 cm breaking wave.

The vertical depth data of oil is dimensionless, where $Z$ is the actual position height, then $Z_{min}$ and $Z_{max}$ are the height above horizontal and the deepest sinking depth of oil respectively. Dimensionless parameter $\alpha = (Z - Z_{min})/(Z_{max} - Z_{min})$ is the normalized vertical height. Then, the reference Froude number is used to calculate the dimensionless numbers of significant wave height and turbulent energy dissipation rate $\varepsilon$, namely: $\varepsilon H/(gH)^{3/2}$, where $H$ is the water depth and $g$ is the gravitational acceleration. The vertical distribution of turbulent energy dissipation rate (Figure 19) shows that the dimensionless spatial distribution of turbulence is mainly divided into three parts. When $T/4 < t < T/2$, turbulence will generate and maintain stability in this region, which is called "turbulence saturation region", and its influencing factors are mainly related to the breaking scale of incoming bubbles during wave breaking. In a certain area, after the turbulence is saturated, the energy loss increases during wave breaking, and the dissipation effect of turbulence will not increase. Its energy is mainly transmitted to the plume generated by the bubble entering the water. When $t > T/2$, the turbulence intensity decreases linearly, which is mainly due to the weakening of turbulence generation and diffusion. When $t > T$, the turbulent energy dissipation rate is basically consistent with the background turbulent flow field. The reason can be simply summarized as the promoting effect of bubbles on the formation of turbulence. The energy carried by the bubble into the water is converted into potential energy. After the bubble enters the water, there are a large number of vortices and small-scale turbulence in the wake. Because the velocity of the bubble at the maximum water entry depth is zero, the turbulent energy dissipation rate at the maximum water entry depth in the image remains at a small level. At the same time, it can be observed that there is strong mixing and strong turbulent diffusion when the bubble sinks and rises, the influence range of turbulence increases, and the influence depth increases. In general, the influence depth of turbulence in the figure is as follows: under the action of breaking waves with a wave height of 15 cm and 25 cm, the oil continues to sink after one cycle of a breaking wave, and it does not reach the maximum depth in one cycle. Under the action of a breaking wave with a wave height of 35 cm, the oil quickly reaches the maximum depth in one cycle. This is because the period becomes larger with the increase of the significant wave height, which buys time for the oil to sink in a period.

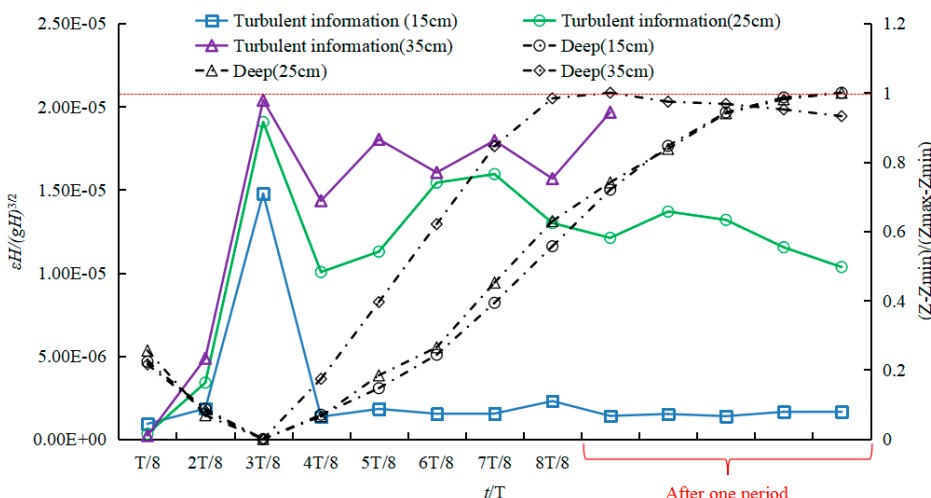

**Figure 19.** Relationship between turbulent dissipation rate and submergence depth under breaking waves.

## 5. Conclusions

Aiming at the application scenarios of detection, artificially synthesized sample training data is used in this paper. The test results showed that the different wave situations of sunken and submerged would not have a great impact on the detection results, indicating it was feasible to use the proposed method to realize the rapid and accurate detection of sunken and submerged oil. This research obtained the following conclusions:

The oil detection method based on multi-strategy fusion is studied to improve the detection accuracy. First, the feature pyramid is used for feature fusion. By adding up sampling and pixel values, the texture features of the image are combined with high-level semantic features, which reduces the position offset caused by multiple sampling of small objects. Then it uses RoIAlign to cancel the quantization operation and uses bilinear interpolation to calculate pixel values, reducing the position information offset caused by the quantization operation in the local feature mapping. Finally, the samples with high confidence are mined, and the entropy is used as the difficulty feedback of the samples, which improves the model's judgment on difficult samples such as small objects. After detection, the use of feature fusion and improved feature mapping can increase detection accuracy by 62.29% and 67.60%, respectively.

On the basis of using synthesized sample training data, this research realized the accurate and fast detection of spilled oil. As the number of synthetic data increases, the detection accuracy gradually increases. Adding 1838 synthesized data in the training set, the average accuracy of the oil detection is increased by 79.72%, which solves the current defects of the difficulty of obtaining oil-moving images and the high cost of image labeling. In addition, the *AP* of FPN + RoIAlign + synthetic data method for labeled oil detection is 18.56% higher than that of oil detection without labeling.

Under the three breaking waves, the liquid oil is hit and sunk to form sunken and submerged oil. The submergence depth for breaking waves in this research shows a good binomial growth trend. For a 15 cm breaking wave, the oil stays under the water for nearly 3.00 s, and the oil reaches the deepest position of 0.181 m. For a 25 cm breaking wave, the oil stays under the water for nearly 3.50 s, and the oil reaches the deepest position of 0.205 m. For a 35 cm breaking wave, the oil stays under the water for nearly 4.20 s, and the oil reaches the deepest position of 0.250 m. Under the action of breaking waves of 15 cm and 25 cm, the oil continues to sink after wave breaking, and it does not reach the maximum depth in one cycle. Under the action of 35 cm breaking wave, the oil quickly reaches the maximum depth in one cycle.

The detection of liquid oil in the breaking wave is a relatively complex target detection problem. If the researchers want to improve its accuracy and achieve better detection results, they still need to continue to optimize this research and improve the method of

target detection. In addition, the process of detection methods based on deep learning is more complicated. The two-stage detection method used in this paper has high accuracy but it consumes too much time. How to further improve the detection speed under the premise of the same accuracy is also an issue that needs attention in the next step.

**Author Contributions:** Conceptualization and writing—original draft preparation, S.F.; validation and funding acquisition, L.M.; methodology, K.L.; validation, D.L. All authors have read and agreed to the published version of the manuscript.

**Funding:** This work was supported by the National Natural Science Foundation of China (Grant No. U2006210), Key Special Project for Introduced Talents Team of Southern Marine Science and Engineering Guangdong Laboratory (Guangzhou) (Grant No. GML2019ZD0604), Shenzhen Fundamental Research Program (Grant No. JCYJ20200109110220482).

**Institutional Review Board Statement:** Not applicable.

**Informed Consent Statement:** Not applicable.

**Data Availability Statement:** The study did not report any data.

**Conflicts of Interest:** The authors declare no conflict of interest.

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
