# Peer review of "Detection of the Sinking State of Liquid Oil in Breaking Waves Based on Synthesized Data: A Behavior Process Study of Sunken and Submerged Oil"

_jmse, doi:10.3390/jmse10050604_

Round 1

Reviewer 1 Report

This is an interesting and useful research. To improve its impact and readability, I suggest that you have it edited for English before I can provide technical review. A few correction/suggestions are in the attached.

Author Response

Response to Reviewer 1 Comments

The authors gratefully acknowledge the efforts of the editor and the reviewers in making positive suggestions and comments for improving the paper. The comments have been appropriately addressed which are detailed as follows. In addition, the revised manuscript has been reviewed by a native English editor.

Point 1: This is an interesting and useful research. To improve its impact and readability, I suggest that you have it edited for English before I can provide technical review. A few correction/suggestions are in the attached.

Response 1: Thanks very much for Reviewer 1’s kind work. On behalf of my co-authors, I would like to express our great appreciation to you. The authors gratefully acknowledge the efforts of the reviewer in making positive suggestions and comments for improving the paper. We have adjusted and revised the full text of the paper. In the revised version of the manuscript, we have highlighted the changes in the manuscript. Then, we have completed the correction/suggestions in the attached. Also we hope the reviewer can accept our paper, thank you so much!

Reviewer 2 Report

The present research paper contributes with methods for the issue of identifying liquid submerged oil pollution. The authors developed a new method to create specific markers for oil that automatically generates labeling files during computer synthesizing. I have the following comments and suggestions for including information and improvements:

Abstract

I suggest that the authors add information about the methods used to obtain the results, even general. The current abstract does not contain information on this aspect, making it a fragile summary.

Introduction

Throughout the introductory paragraphs, the authors should illustrate more about the state of the art in detecting the sinking state of liquid oil in breaking waves based on synthesized data. More citations should be included throughout the introduction, especially between lines 71 - 125.

Another critical point that will bring more information to the reader is to describe directly and with greater clarity the objectives of the work. Of course, when reading the complete introduction, we know what the objective is. However, making it more explicit will help readers better understand the work's importance.

The novelty of this research

In the section "The novelty of this research," it is imperative that the authors cite references that support the readers of how innovative the proposed methods are. It is crucial to guide readers with citations that evidence the theoretical, conceptual alignment of how robust and innovative the proposed paper is in terms of methods. One way to facilitate this path is to cite a sequence of articles that show the evolution of this scientific field.

Methods / Results

More information about the model used as a classifier, including which classifier family is associated with the deep learning model used, classifier settings, and parameterization (e.g., learning rate, number of neurons, hidden layers, maximum iteration, momentum), as well as information about the sample design for training the classifier and feature space used, must be include.

The authors must include more information about accuracy assessment in the methods section. The errors matrix and the omission and commission errors per class must be inserted too.

Disscution

Citations that give conceptual support to the results found in this section must be included. It was impossible to view any citations in the current version of the paper. This weakens the article, so it must be done. It is also imperative that the authors include a topic that compares the results found from the methods and analyses proposed in the article with the literature results.

Conclusions

In this section, I suggest that more information be added about future advances in this field of research, based on the results found in this article.

Round 2

Reviewer 2 Report

Thanks for the authors made efforts to revise the manuscript. The manuscript can be published in the current form in the Journal.